# A Study on the Relationship between Pragmatic Language Development and Socioeconomic Status in Arab Preschoolers with and without Pragmatic Language Impairment

Fawaz Qasem [1], Ahmed Alduais [2,*], Hind Alfadda [3,*], Najla Alfadda [4] and Lujain Al Amri [5]

[1] Department of English, University of Bisha, Bisha 67714, Saudi Arabia; faqasem@ub.edu.sa
[2] Department of Human Sciences, University of Verona, 37129 Verona, Italy
[3] Department of Curriculum and Instruction, King Saud University, Riyadh 11451, Saudi Arabia
[4] Department of English Language and Translation, King Saud University, Riyadh 11451, Saudi Arabia; nalfadda@ksu.edu.sa
[5] Speech Pathology Division, Jeddah Institute for Speech and Hearing and Medical Rehabilitation, Jeddah 23432, Saudi Arabia; lujain@jish.med.sa
* Correspondence: ahmedmohammedsaleh.alduais_02@univr.it (A.A.); halfadda@ksu.edu.sa (H.A.)

**Abstract:** The scope of this study is threefold. First, it examines socioeconomic status (SES) and pragmatic language development (PLD), hypothesizing that parental education and employment levels are mediators, where SES affects PLD mainly through its influence on parental education and employment levels. Second, we used quantitative (age) and qualitative (gender) variables as moderators, hypothesizing that gender and age moderate the relationship between classroom interaction (CI), social interaction (SI), and personal interaction (PI) skills and level of PLD. Third, characteristics of PLD in preschoolers with and without pragmatic language impairment (PLI) are compared. The Arabic Pragmatic Language Skills Inventory (A-PLSI) was administered by preschool teachers and speech-language pathologists in Saudi Arabia to 264 preschoolers with and without neurodevelopmental disorders to assess their PLD. Additionally, the SES of the parents of the same number of participants was surveyed. Results show that the more CI, SI, and PI a preschool child has, the greater the likelihood of typical PLD, regardless of the parents' employment or education. Further, we obtained evidence that CI, SI, and PI all grow together with age. However, these three elements (namely, CI, SI, and PI) do not seem to be moderated or altered by gender. Typical PLD may be attainable when preschool children demonstrate typical mental and physical development, in contrast to children with psychiatric histories who display atypical PLD. These findings suggest that preschool children with more CI, SI, and PI will exhibit more typical PLD. The achievement of this goal results in a sustainable society for children.

**Keywords:** pragmatic language development; pragmatic language impairment; preschoolers; socioeconomic status; Arabic Pragmatic Language Skills Inventory; Arabic; Saudi Arabia

## 1. Introduction

Pragmatic language development (PLD) refers to children's communication skills both linguistic and non-linguistic, which include several influences such as socialization by caregivers, fathers and siblings, teachers, peers, cognition, knowledge, and effort [1]. A previous study in the Arabian context with children whose mother tongue language is Arabic with and without neurodevelopmental disorders reported that "PLI severity seems to be controlled by the primary disorder type: congenital (developmental-dysphasia), biolinguistic (genetic) or neurolinguistic (acquired child aphasia)" [2]. Research on the same populations also reported differences in PLI according to assessment type and whether the participants were in school or clinical settings [2–5]. More importantly, researchers found that there no relationship between PLD and preschool education [6,7].

In the present study, the researchers attempted to examine the SES and PLD of Arabic-speaking preschool children with and without neurodevelopmental disorders. They argue that SES, including the parental education and employment levels, affects PLD. The study also attempts to statistically investigate how age and gender act as moderators between classroom interaction (CI), social interaction (SI), and personal interaction (PI) skills and level of PLD. Since there is a shortage of studies addressing PLD and SES in the Arabian context, this study presents the characteristics of PLD in preschoolers with and without pragmatic language impairment in relation to SES, including parents' employment and education.

Beyond the traditional linguistic concern of grammar and meaning, pragmatics looks at acquisition and the ability to know when, where, and how to speak and express communicative intentions according to context [8–10]. Therefore, social interactions and communications in various contexts play a major role in child language development, including pragmatic development skills. Socioeconomic status (SES) has positive or negative implications for pragmatic conversational and interactional skills in the early stages of pragmatic development of children's speech. Based on PLD assessment tools, several studies in various languages have indicated the influences of the social surrounding factors on the PLD of children's speech such as parental education and employment levels. These factors affect children with and without PLI. In this study, the researchers look at the influence of socioeconomic factors on pragmatic development of Arabic-speaking preschool children. SES, including parent education and income, has a relationship with children's language development. It was found that the differences in the social–pragmatic aspects of speech of children are apparent. Parents with a high SES verbally encourage and provide affirmation to their children more than parents with lower SES. The parents with low SES verbally discourage their children's behavior more than the parents with high SES [11].

Several studies have shown that parents with high income use long sentences and sentences with various structures and words [12,13]. A similar study [14] showed that mothers with high SES use longer utterances with more different words when they talk to their children than mothers with low SES, and their children have larger vocabularies. However, SES has a very small effect on children's comprehension in pragmatics, and it was found in some studies that there is no relationship between communication skills and the understanding of pragmatic abilities and aspects [15–17].

SES also influences children with language impairment, not only children with typical language development. Wild [18] found that there was an early influence of SES on language outcome in a cohort matched for biomedical risk, suggesting that very early language interventions may be required for low-SES preterm toddlers. Similarly, (Betancourt, Brodsky, and Hurt [19] examined the effect of SES on infant language at 7 months of age and the relationship between maternal vocabulary skills and infant language function. It was found that infants with low SES performed less well than those with high SES on language skills. Mitchell et al. [20] explored whether toddlers and preschoolers with autism spectrum disorder (ASD) with low SES are more likely to experience language delays by examining particular expressive language (EL) and receptive language (RL) skills. The results demonstrated that variability in EL and RL skills in young children with autism can be accounted for by socioeconomic variables.

Previous literature reported that social communication affects PLD. Comparing the pragmatic skills of non-maltreated children to maltreated children, several studies indicated that the non-maltreated children made fewer utterances about a given subject, and their comments were fewer than their peers of the same age [21,22]. This was due to less social communication exposure. For instance, Coster [21] examined the pragmatic skills of 20 maltreated children, including neglected children, aged 30–33 months and 29 days. Children's daily and social communications with their mother were analyzed to study two pragmatic factors, including (a) the production of second-language intentional communications (e.g., giving names to things, asking for things, describing objects) and (b) the ability to continue and maintain the flow of conversations. The results indicated that

maltreated children expressed fewer communication intentions, including fewer requests to their mothers and fewer comments about the objects in their setting. In comparison to previous research, recent research has also indicated that preschool children are vulnerable to several risk factors (hearing impairment [23], preterm children [24], emotional competence [25]) leading to atypical language development, including PLD [26]. Another study concluded that pragmatic language skills in preschool children predict psychosocial and quality-of-life outcomes [27]. Another study conducted on preschool children with low SES in Italy showed that early intervention and training of preschool children can help bridge the gap between children with different SES to enhance literacy and oral language skills [12]. This is consistent with findings from another study in Italy highlighting the importance of narrative listening to foster communication skills for preschoolers [28]. More interestingly, another recent study reported that SES is independently related to cognitive and logistic skills in preschool children [29].

As a result, the present study attempted to answer the following three questions. (1) Does SES (parental education and employment levels) mediate PLD in preschoolers with and without neurodevelopmental disorders? (2) Do age and gender moderate the relationship between CI, SI, and PI skills and level of PLD? (3) What are the characteristics of preschoolers with and without PLI with regard to their age, gender, and SES differences?

## 2. Method

### 2.1. Sample

The study's theoretical population consists of preschoolers who speak Arabic as their mother tongue with or without psychiatric histories. Children in Saudi Arabia with and without psychiatric histories made up the accessible population. Samples of preschoolers in Saudi Arabia included those who were enrolled or not enrolled in preschools. Children who have not started basic education were considered preschoolers. These could be ≤7.0 years. A total of 237 preschoolers without pragmatic language impairment and 27 preschoolers with pragmatic language impairment were analyzed. Additional information about this population can be found in Table 1.

**Table 1.** Respondents' Characteristics.

|  | School Setting (N) | Clinical Setting (N) | % | |
|---|---|---|---|---|
| **Age Group** | 237 | 27 | | |
| 4 | 14 | 15 | 6 | 55 |
| 5 | 19 | 8 | 8 | 30 |
| 6 | 56 | 40 | 24 | 150 |
| 7 | 148 | 0 | 62 | 0 |
| **Gender Group** | | | | |
| Female | 142 | 5 | 60 | 19 |
| Male | 95 | 22 | 40 | 81 |
| **City Group** | | | | |
| Riyadh | 158 | | 67 | |
| Eastern region | 18 | | 8 | |
| Jeddah | 14 | 27 | 6 | 100 |
| Khamis Mushait | 14 | | 6 | |
| Makkah | 10 | | 4 | |
| Other cities | 23 | | 9 | |

**Table 1.** *Cont.*

| | School Setting (N) | Clinical Setting (N) | % | |
|---|---|---|---|---|
| **Socioeconomic Status** | | | | |
| *Father employment* | | | | |
| Employed | 227 | 23 | 96 | 85 |
| Unemployed | 10 | 4 | 4 | 15 |
| *Mother employment* | | | | |
| Employed | 127 | 2 | 54 | 7 |
| Unemployed | 110 | 25 | 46 | 93 |
| *Father education* | | | | |
| Middle school | 6 | 0 | 3 | 0 |
| Secondary school | 58 | 7 | 24 | 26 |
| Bachelor's degree | 141 | 13 | 59 | 48 |
| Master's degree | 17 | 4 | 7 | 15 |
| Doctorate | 15 | 3 | 6 | 11 |
| *Mother education* | | | | |
| Middle school | 15 | 1 | 6 | 4 |
| Secondary school | 44 | 7 | 19 | 26 |
| Bachelor's degree | 150 | 16 | 63 | 59 |
| Master's degree | 23 | 2 | 10 | 7 |
| Doctorate | 5 | 1 | 2 | 4 |
| **Exceptionality Status** | | | | |
| No exceptionality | 237 | 27 | NA | NA |
| Attention-deficit hyperactivity disorder | NA | 3 | 11 | NA |
| Hearing impairment | NA | 3 | 11 | NA |
| (Speech and) language delay | NA | 10 | 37 | NA |
| Childhood apraxia/dyslexia | NA | 5 | 18.5 | NA |
| Autism spectrum disorder | NA | 4 | 15 | NA |
| Developmental delay | NA | 1 | 3.5 | NA |
| Down's syndrome | NA | 1 | 3.5 | NA |

For the school setting group, a total of 237 preschoolers aged 4 to 7 years, both boys and girls, from various areas of Saudi Arabia were randomly selected for participation in this study. The participants are described in detail in Table 1. For the clinical setting group, we selected 27 Arabic-speaking Saudi children with various communication needs for assessment at the Jeddah Institute for Speech and Hearing and Medical Rehabilitation (JISH), Jeddah, Saudi Arabia. JISH specializes in the assessment and treatment of children and adults with various communication disorders. The participants consisted of twenty children with different neurodevelopmental disorders and seven with no concomitant disorders. Each parent of the children participating in the study signed an informed consent form. This study was also approved by the JISH Research Committee. Table 1 summarizes the characteristics of participants in a clinical setting.

*2.2. Instrument*

The A-PLSI was normed on 264 preschool children with and without PLI. The authors reported representative normative information and high construct validity, criterion-related validity, and internal consistent reliability for the validated version. The validated version was approved to identify children with and without PLI, document progress of PLD, and determine strengths and weaknesses in pragmatic language skills.

According to Gilliam and Miller [23], the PLSI is an instrument designed to evaluate children's pragmatic language skills. The instrument is theoretically designed on the basis of pragmatics [24–26]. The authors of the instruments adopted the rules of communication introduced by Bates [25]. These include: (1) cooperating with your conversational partner, (2) telling the truth, (3) considering the four maxims of speech (quality, quantity, relevance,

and manner), (4) requesting only relevant information, (5) providing adequate background information, (6) being attentive, and (7) adapting the language to the situation [23].

PLSI consists of 45 items divided into three subscales: classroom interaction (CI), social interaction (SI), and personal interaction (PI). Among the uses of the instrument are: (1) identifying students with pragmatic language impairment, (2) documenting their progress in pragmatic language ability, (3) identifying strengths and weaknesses in pragmatic language ability, and (4) data collection for research [23].

The English PLSI was normed on 1175 children ages five to twelve from different areas within the United States. The data were collected between 2001 and 2004. The authors note that they included data on children with disabilities as well. This instrument has been evaluated for alpha coefficients, test-retest reliability, and inter-rater reliability. Moreover, content validity, item discrimination, and criterion-related validity were also reported. Other aspects of validity included construct validity and factor analysis. All three subscales and the pragmatic language index were found to be satisfactory, confirming the instrument's suitability as a measure.

*2.3. Design*

Two comparison groups are used in the study. The first group is a school setting, while the second is a clinical setting. Since the purpose of the study was to examine the possible effect of SES on PLD, it was important to consider comparing data from children with and without a history of psychiatric illness. Data from both groups of participants were randomly selected. Participants in the clinical group were selected based on age, but no limitations were placed on the type of disorder or even IQ level. The inclusion of participants with disabilities was intended to uncover differences between parental SES in employment and education, which may be influencing the level of support provided to their children.

*2.4. Procedures*

Data were collected between 19 October 2021 and 13 January 2022. Preschool teachers administered the instrument in randomly selected schools in Saudi Arabia (see Table 1). For each participant, the administration time ranged between five minutes and ten minutes. The preschool teachers were trained to administer the test by the third and fourth authors, who had been trained by the first author. Teachers filled out the required information based on their knowledge and experience, as well as their time spent with the students. To collect data from preschools, approval was obtained from an institutional review board (IRB) at King Saud University, Saudi Arabia. In this study, no participants were enrolled in basic education regardless of their age at the time of data collection.

Regardless of the severity or nature of the communication disorder or the length of time spent in therapy, participants who met the criteria were included in the study. After the participants had been selected, the speech-language pathologist (SLP) administered the test and provided therapy. Each patient's treatment plan included goals related to pragmatics and social skills. However, treatment plans that catered to children who have a problem with social skills or are diagnosed with ASD included more goals that target specific weaknesses. To achieve these goals, various methods were employed, such as social scripts, social stories, and social groups to generalize skills. All treatment plans were family-oriented, which meant that parents were also an integral part of therapy. To transfer learned skills to the home environment, the parents were included in the therapy sessions.

Several steps were involved in the analysis of the data. Initially, all the data were transferred from booklets to Excel sheets. Excel sheets were checked for accuracy. After this, the Excel spread sheets were translated into English, as the original ones were created in Arabic. The collected data were analyzed using Minitab 18 and Jamovi 2.2.2. Both descriptive and inferential tools were used to analyze the collected data and reach the study's objective. Detailed results are presented in the following section.

### 3. Results

This study examined the relationship between PLD and CI, SI, and PI, as well as SES as a mediating factor. Gender and age were also examined as moderating factors. Further, it compared the characteristics of PLD between children with and without a history of mental illness. This section presents three themes: mediation analysis, moderation analysis, and preschoolers' performance on PLD with or without PLI. This study included 264 preschoolers (M = 6.24, SD = 1.02). The sample consisted of two groups: schools (N = 237; mean = 6.43, standard deviation = 0.873) and clinical settings (N = 27; mean = 4.59, standard deviation = 0.747).

*3.1. Socioeconomic Status and Pragmatic Language Development*

A simple mediation analysis was conducted to assess if socioeconomic status mediated the relationship between CI, SI, and PI and PLD using Jamovi 2.2.2. To determine whether a mediating relationship was supported by the data, 12 regressions were conducted. For mediation to be supported, four terms must be met: (1) the independent variables must be related to the dependent variable, (2) the independent variables must be related to the mediator variables, (3) the mediator must be related to the dependent variables while in the presence of the independent variables, and (4) the independent variables should no longer be significant predictors of the dependent variable in the presence of the mediator variables. In the following analyses, the independent variables were CI, SI, and PI; the dependent variable was PLD; and the mediator variables were father employment (FE), mother employment (ME), father education (FED), and mother education (MED).

*3.2. Classroom Interaction, Pragmatic Language Development, and Parental Employment*

The first mediation analysis was performed to assess the mediating effect of father employment on the linkage between CI and PLD. The results (see Table 2 and Figure 1) reveal that the total effect of CI on PLD was significant (H1: β = 2.529, t = 52.39, *p* < 0.001). With the inclusion of the mediating variable (FE), the impact of CI on PLD was still found significant (β = 2.525, t = 52.349, *p* < 0.001). However, the indirect effect of CI on PLD through FE was found insignificant (β = 0.004, t = 0.822, *p* < 0.411) (see Figure 2). These results indicate that the first criterion for mediation was satisfied but not the last three criteria. Given this, the relationship between CI and PLD is not mediated by FE, and hence mediation is not approved.

**Table 2.** Mediation Analysis for Classroom Interaction, PLD, and Father Employment.

| Effect and Path | Label | Estimate | SE | Z | *p* | Mediation % |
|---|---|---|---|---|---|---|
| Indirect | a × b | −0.00438 | 0.00533 | 0.00533 | −0.822 | 0.173 |
| Direct | c | 2.52518 | 0.04824 | 0.04824 | 52.349 | 99.827 |
| Total | c + a × b | 2.52080 | 0.04812 | 0.04812 | 52.389 | 100.000 |
| CI→FE | a | −7.43−4 | 4.914 | −1.513 | 0.130 | |
| FE→PLD | b | 5.90 | 6.0221 | 0.980 | 0.327 | |
| CI→PLD | c | 2.53 | 0.0482 | 52.349 | <0 .001 | |

The second mediation analysis was performed to assess the mediating effect of ME on the linkage between CI and PLD. The results (see Table 3 and Figure 3) reveal that the total effect of CI on PLD was significant (H1: β = 2.5208, t = 52.39, *p* < 0.001). With the inclusion of the mediating variable (ME), the impact of CI on PLD was still found significant (β = 2.4941, t = 49.35, *p* < 0.001). However, the indirect effect of CI on PLD through ME was found insignificant (β = 0.026, t = 0.822, *p* < 0.115) (see Figure 4). These results indicate that the first two criteria for mediation were satisfied but not the last two criteria. Given this, the relationship between CI and PLD is not mediated by ME, and hence mediation is not approved.

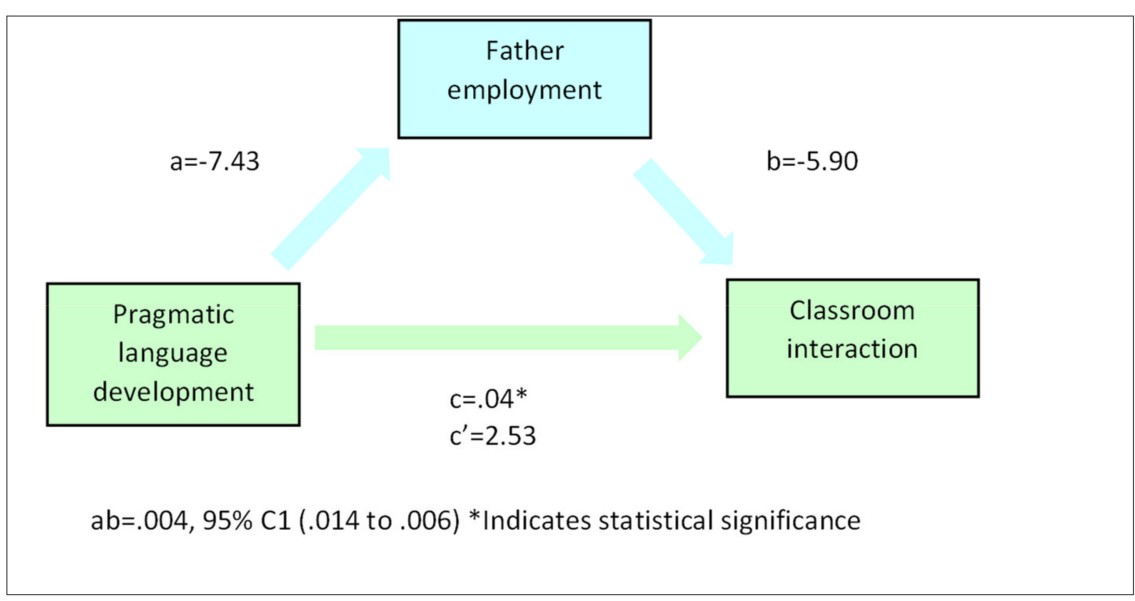

**Figure 1.** Mediation Illustration for Classroom Interaction, PLD, and Father Employment.

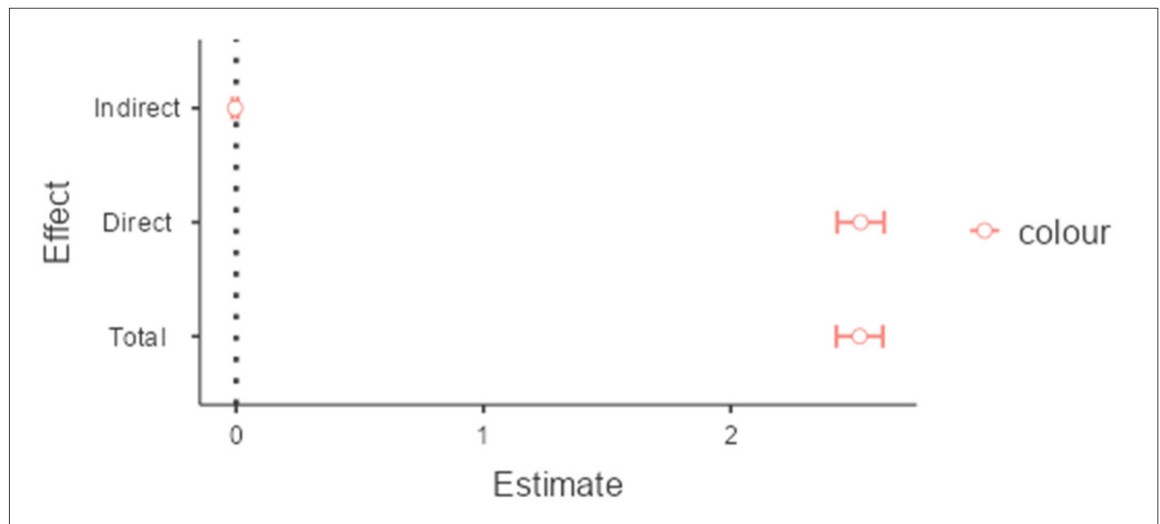

**Figure 2.** Mediation Effect for Classroom Interaction, PLD, and Father Employment.

**Table 3.** Mediation Analysis for Classroom Interaction, PLD, and Mother Employment.

| Effect and Path | Label | Estimate | SE | Z | p | Mediation % |
|---|---|---|---|---|---|---|
| Indirect | a × b | 0.0267 | 0.0169 | 1.58 | 0.115 | 1.06 |
| Direct | c | 2.4941 | 0.0505 | 49.35 | <0.001 | 98.94 |
| Total | c + a × b | 2.5208 | 0.0481 | 52.39 | <0.001 | 100.00 |
| CI→ME | a | −0.00573 | 0.00104 | −5.50 | <0.001 | |
| ME→PLD | b | −4.65852 | 2.82874 | −1.65 | 0.100 | |
| CI→PLD | c | 2.49410 | 0.05054 | 49.35 | <0.001 | |

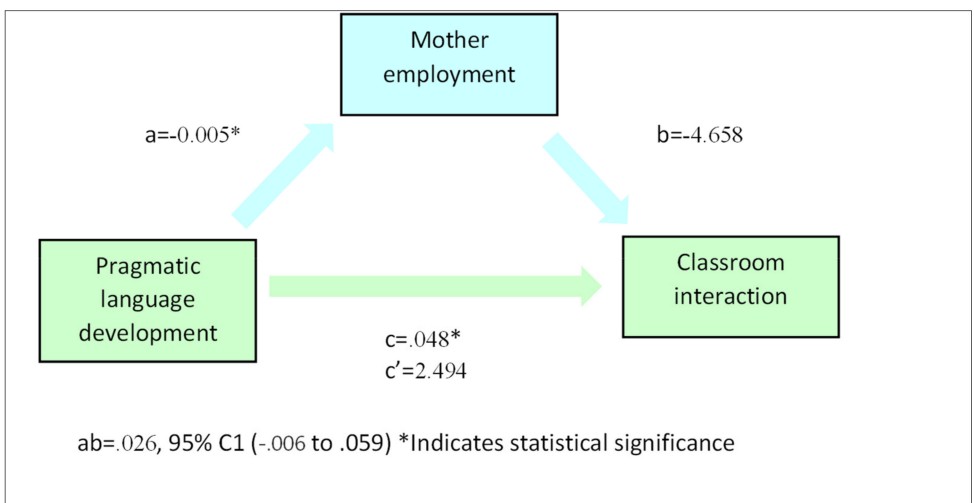

**Figure 3.** Mediation Illustration for Classroom Interaction, PLD, and Mother Employment.

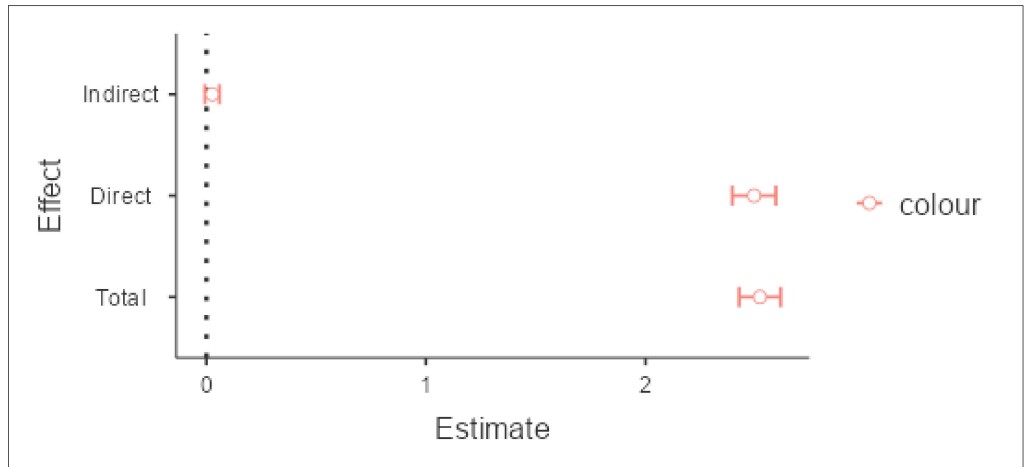

**Figure 4.** Mediation Effect for Classroom Interaction, PLD, and Mother Employment.

*3.3. Social Interaction, Pragmatic Language Development, and Parental Employment*

The third mediation analysis was performed to assess the mediating effect of FED on the linkage between CI and PLD. The results (see Table 4 and Figure 5) reveal that the total effect of CI on PLD was significant (H1: $\beta = 2.52$, t = 52.39, $p < 0.001$). With the inclusion of the mediating variable (FED), the impact of CI on PLD was still found significant ($\beta = 2.52$, t = 52.420, $p < 0.001$). However, the indirect effect of CI on PLD through FED was found insignificant ($\beta = -1.494$, t = 0.822, $p < 0.927$) (see Figure 6). These results indicate that the first criterion for mediation was satisfied but not the last three criteria. Given this, the relationship between CI and PLD is not mediated by FED, and hence mediation is not approved.

**Table 4.** Mediation Analysis for Classroom Interaction, PLD, and Father Education.

| Effect and Path | Label | Estimate | SE | Z | p | Mediation % |
|---|---|---|---|---|---|---|
| Indirect | a × b | −1.494 | 0.00163 | −0.0912 | 0.927 | 0.00590 |
| Direct | c | 2.52 | 0.04809 | 52.4206 | <0.001 | 99.99410 |
| Total | c + a × b | 2.52 | 0.04812 | 52.3891 | <0.001 | 100.00000 |
| CI→FED | a | −3.464 | 0.00374 | −0.0925 | 0.926 | |
| FED→PLD | b | 0.430 | 0.79127 | 0.5431 | 0.587 | |
| CI→PLD | c | 2.521 | 0.04809 | 52.4206 | <0.001 | |

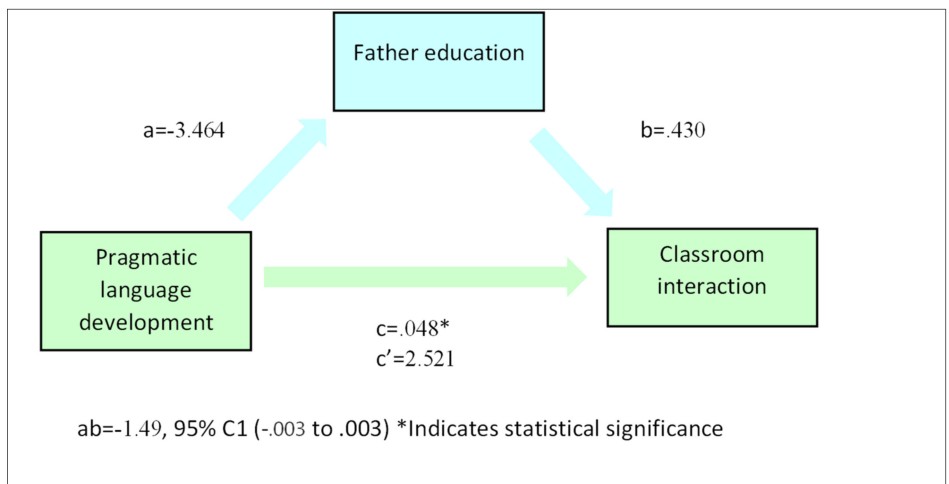

**Figure 5.** Mediation Illustration for Classroom Interaction, PLD, and Father Education.

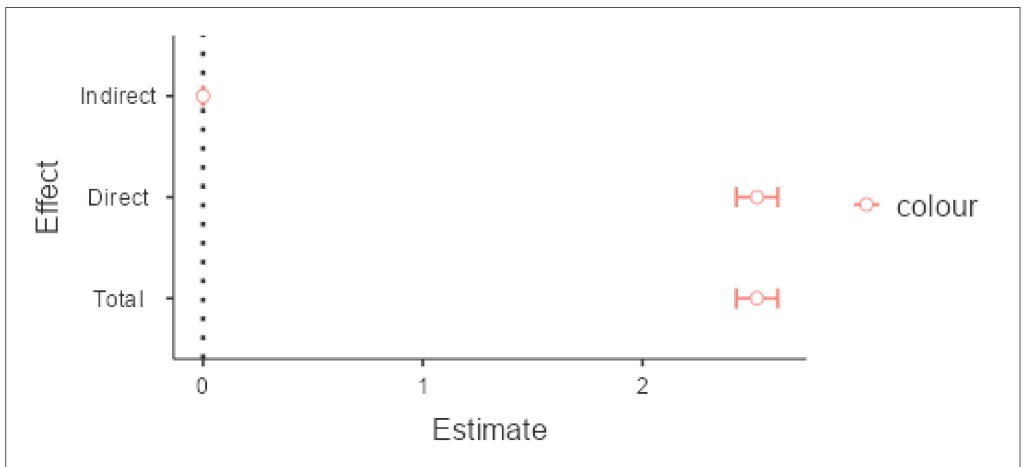

**Figure 6.** Mediation Effect for Classroom Interaction, PLD, and Father Education.

The fourth mediation analysis was performed to assess the mediating effect of MED on the linkage between CI and PLD. The results (see Table 5 and Figure 7) reveal that the total effect of CI on PLD was significant (H1: β = 2.52, t = 52.39, *p* < 0.001). With the inclusion of the mediating variable (MED), the impact of CI on PLD was still found significant (β = 2.52, t = 52.41, *p* < 0.001). However, the indirect effect of CI on PLD through MED was found insignificant (β = −0.001, t = 0.410, *p* < 0.681) (see Figure 8). These results indicate that the first criterion for mediation was satisfied but not the last three criteria. Given this, the relationship between CI and PLD is not mediated by MED, and hence mediation is not approved.

**Table 5.** Mediation Analysis for Classroom Interaction, PLD, and Mother Education.

| Effect and Path | Label | Estimate | SE | Z | *p* | Mediation % |
|---|---|---|---|---|---|---|
| Indirect | a × b | 0.00117 | 0.00284 | 0.410 | 0.681 | 0.0463 |
| Direct | c | 2.51963 | 0.04807 | 52.413 | <0.001 | 99.9537 |
| Total | c + a × b | 2.52080 | 0.04812 | 52.389 | <0.001 | 100.0000 |
| CI→MED | a | −0.00169 | 0.00360 | −0.471 | 0.638 | |
| MED→PLD | b | −0.68910 | 0.82204 | −0.838 | 0.402 | |
| CI→PLD | c | 2.51963 | 0.04807 | 52.413 | <0.001 | |

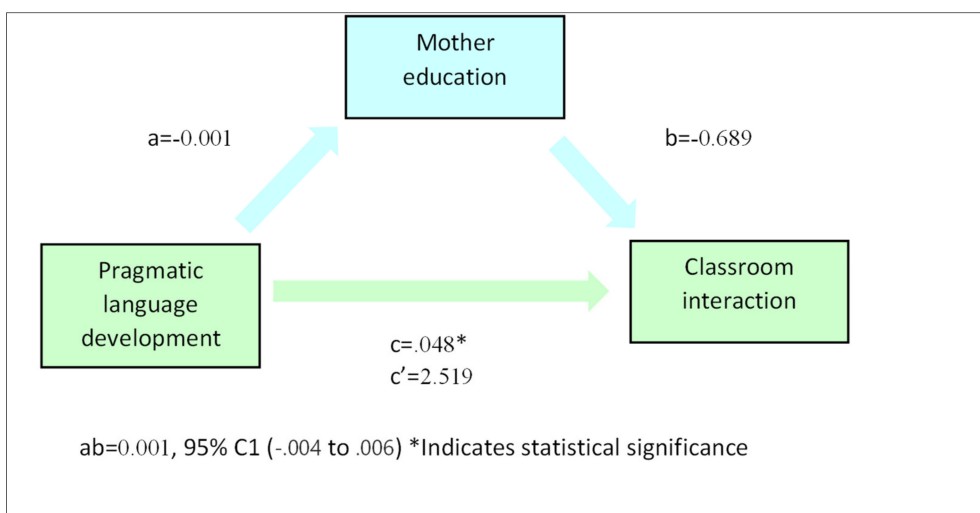

**Figure 7.** Mediation Illustration for Classroom Interaction, PLD, and Mother Education.

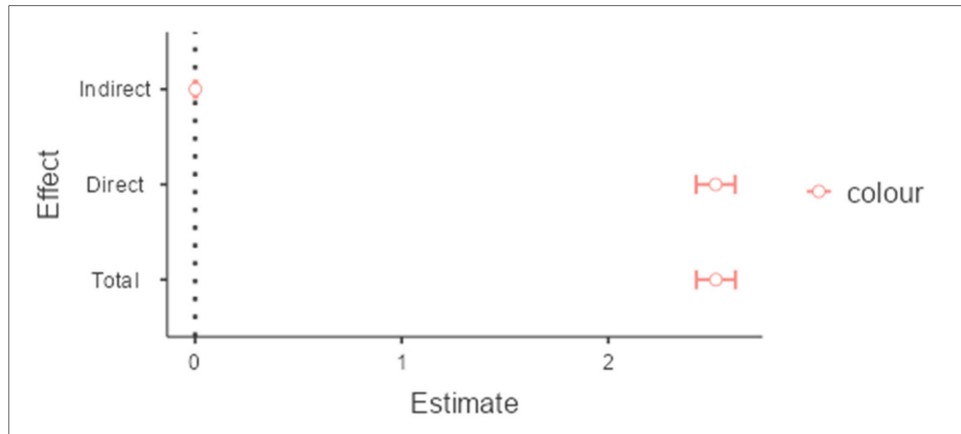

**Figure 8.** Mediation Effect for Classroom Interaction, PLD, and Mother Education.

### 3.4. Social Interaction, Pragmatic Language Development, and Parental Employment

The fifth mediation analysis was performed to assess the mediating effect of FE on the linkage between SI and PLD. The results (see Table 6 and Figure 9) reveal that the total effect of SI on PLD was significant (H1: $\beta$ = 2.72, t = 50.750, $p$ < 0.001). With the inclusion of the mediating variable (FE), the impact of SI on PLD was still found significant ($\beta$ = 2.72, t = 50.901, $p$ < 0.001). However, the indirect effect of SI on PLD through FE was found insignificant ($\beta$ = 0.003, t = 0.629, $p$ < 0.529) (see Figure 10). These results indicate that the first criterion for mediation was satisfied but not the last three criteria. Given this, the relationship between SI and PLD is not mediated by FE, and hence mediation is not approved.

**Table 6.** Mediation Analysis for Social Interaction, PLD, and Father Employment.

| Effect and Path | Label | Estimate | SE | Z | $p$ | Mediation % |
|---|---|---|---|---|---|---|
| Indirect | a × b | 0.00371 | 0.00590 | 0.629 | 0.529 | 0.136 |
| Direct | c | 2.71935 | 0.05342 | 50.901 | <0.001 | 99.864 |
| Total | c + a × b | 2.72306 | 0.05366 | 50.750 | <0.001 | 100.000 |
| SI→FE | a | −3.634 | 0.00360 | −0.680 | 0.497 | |
| FE→PLD | b | −10.24 | 0.82204 | −1.662 | 0.096 | |
| SI→PLD | c | 2.72 | 0.04807 | 50.901 | <0.001 | |

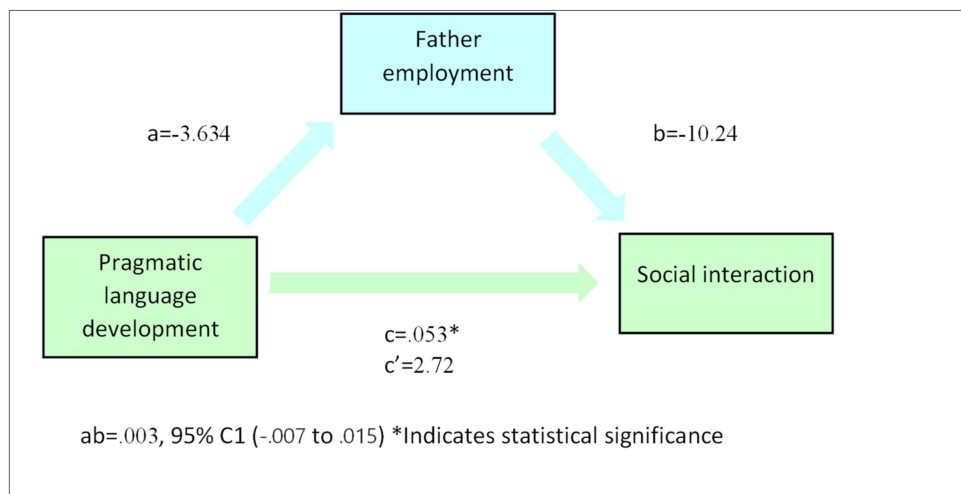

**Figure 9.** Mediation Illustration for Social Interaction, PLD, and Father Employment.

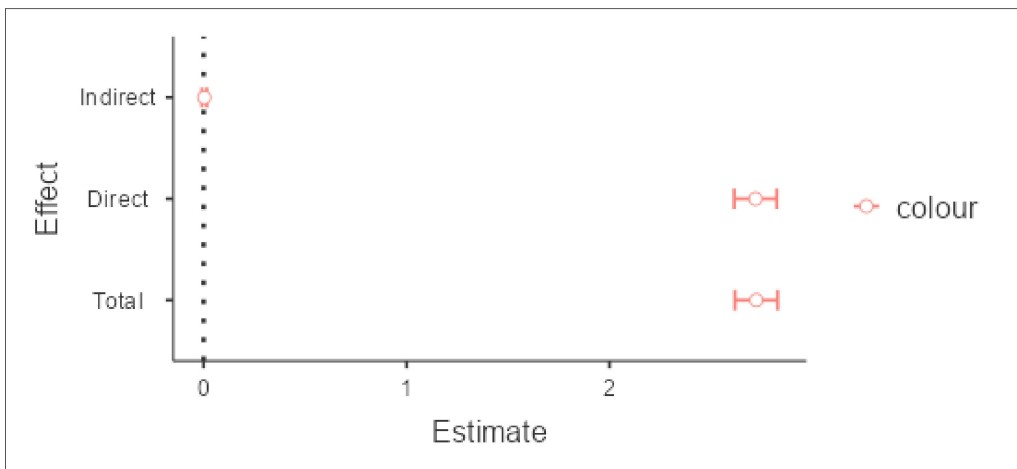

**Figure 10.** Mediation Effect for Social Interaction, PLD, and Father Employment.

The sixth mediation analysis was performed to assess the mediating effect of ME on the linkage between SI and PLD. The results (see Table 7 and Figure 11) reveal that the total effect of SI on PLD was significant (H1: β = 2.72, t = 50.750, $p < 0.001$). With the inclusion of the mediating variable (ME), the impact of SI on PLD was still found significant (β = 2.70, t = 47.606, $p < 0.001$). However, the indirect effect of SI on PLD through ME was found insignificant (β = 0.018, t = 0.944, $p < 0.345$) (see Figure 12). These results indicate that the first two criteria for mediation were satisfied but not the last two criteria. Given this, the relationship between SI and PLD is not mediated by ME, and hence mediation is not approved.

**Table 7.** Mediation Analysis for Social Interaction, PLD, and Mother Employment.

| Effect and Path | Label | Estimate | SE | Z | *p* | Mediation % |
|---|---|---|---|---|---|---|
| Indirect | a × b | 0.0181 | 0.0192 | 0.944 | 0.345 | 0.666 |
| Direct | c | 2.7049 | 0.0568 | 47.606 | <0.001 | 99.334 |
| Total | c + a × b | 2.7231 | 0.0537 | 50.750 | <0.001 | 100.000 |
| SI→ME | a | −0.00646 | 0.00112 | −5.751 | <0.001 | |
| ME→PLD | b | −2.80951 | 2.93540 | −0.957 | 0.339 | |
| SI→PD | c | 2.70491 | 0.05682 | 47.606 | <0.001 | |

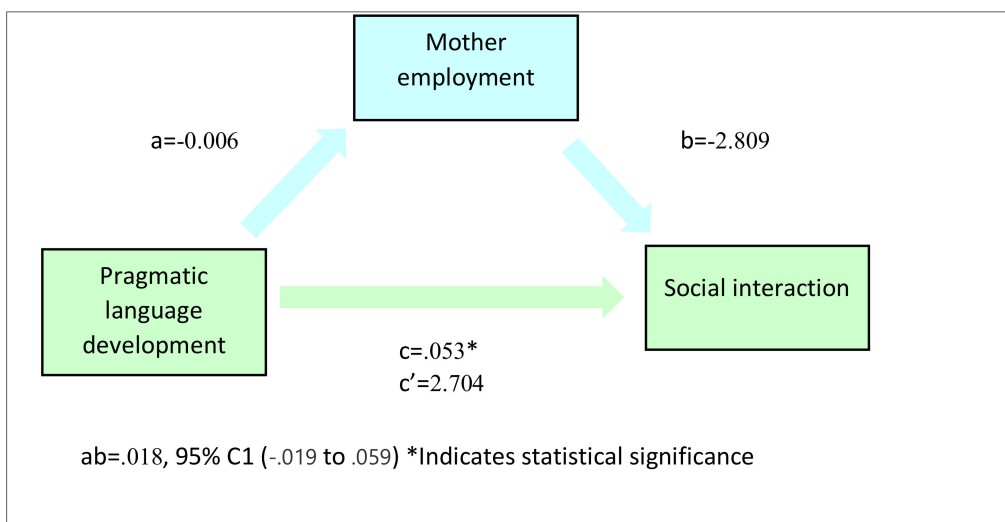

**Figure 11.** Mediation Illustration for Social Interaction, PLD, and Mother Employment.

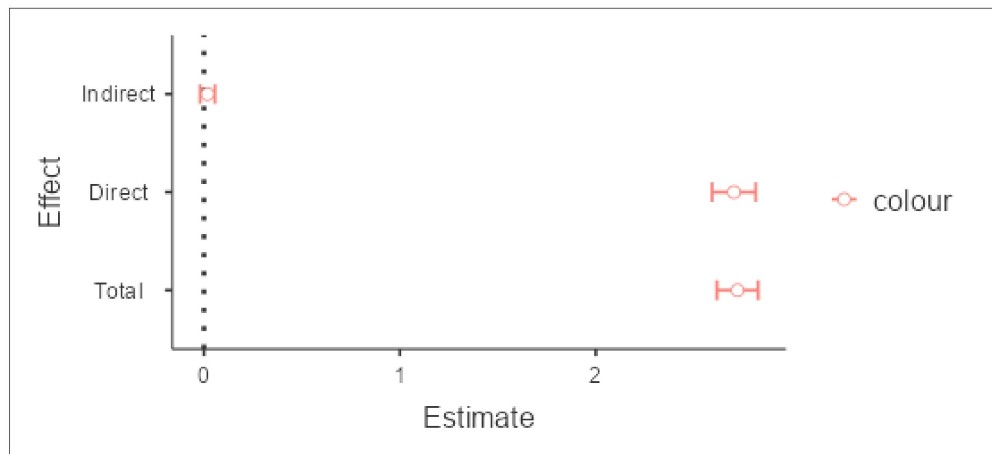

**Figure 12.** Mediation Effect for Social Interaction, PLD, and Mother Employment.

*3.5. Social Interaction, Pragmatic Language Development, and Parental Education*

The seventh mediation analysis was performed to assess the mediating effect of FED on the linkage between SI and PLD. The results (see Table 8 and Figure 13) reveal that the total effect of SI on PLD was significant (H1: $\beta$ = 2.72, t = 50.750, $p$ < 0.001). With the inclusion of the mediating variable (FED), the impact of SI on PLD was still found significant ($\beta$ = 2.72, t = 51.042, $p$ < 0.001). However, the indirect effect of SI on PLD through FED was found insignificant ($\beta$ = −0.003, t = −0.570, $p$ < 0.569) (see Figure 14). These results indicate that the first criterion for mediation was satisfied but not the last three criteria. Given this, the relationship between SI and PLD is not mediated by FED, and hence mediation is not approved.

**Table 8.** Mediation Analysis for Social Interaction, PLD, and Father Education.

| Effect and Path | Label | Estimate | SE | Z | $p$ | Mediation % |
|---|---|---|---|---|---|---|
| Indirect | a × b | −0.00331 | 0.00582 | −0.570 | 0.569 | 0.121 |
| Direct | c | 2.72637 | 0.05341 | 51.042 | <0.001 | 99.879 |
| Total | c + a × b | 2.72306 | 0.05366 | 50.750 | <0.001 | 100.000 |
| SI→FED | a | 0.00245 | 0.00405 | 0.606 | 0.544 | |
| FED→PLD | b | −1.35031 | 0.81124 | −1.664 | 0.096 | |
| SI→PLD | c | 2.72637 | 0.05341 | 51.042 | <0.001 | |

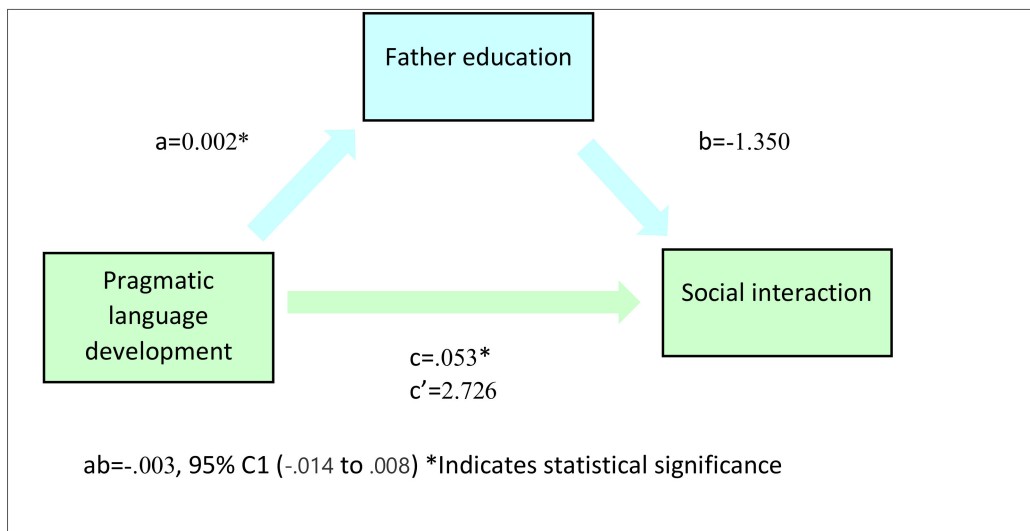

**Figure 13.** Mediation Illustration for Social Interaction, PLD, and Father Education.

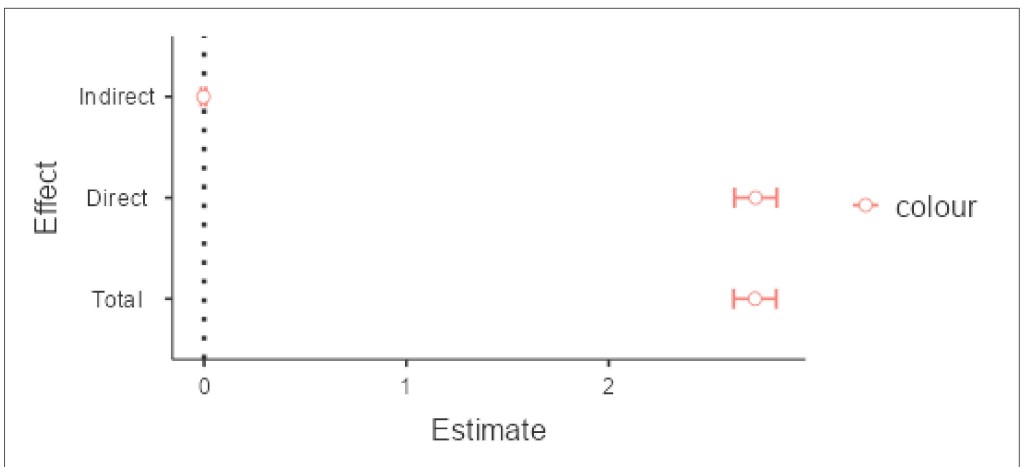

**Figure 14.** Mediation Effect for Social Interaction, PLD, and Father Education.

The eighth mediation analysis was performed to assess the mediating effect of MED on the linkage between SI and PLD. The results (see Table 9 and Figure 15) reveal that the total effect of SI on PLD was significant (H1: $\beta = 2.72$, t = 50.750, $p < 0.001$). With the inclusion of the mediating variable (MED), the impact of SI on PLD was still found significant ($\beta = 2.72$, t = 51.71, $p < 0.001$). However, the indirect effect of SI on PLD through MED was found insignificant ($\beta = 7.594$, t = 0.320, $p < 0.749$) (see Figure 16). These results indicate that the first criterion for mediation was satisfied but not the last three criteria. Given this, the relationship between SI and PLD is not mediated by MED, and hence mediation is not approved.

**Table 9.** Mediation Analysis for Social Interaction, PLD, and Mother Education.

| Effect and Path | Label | Estimate | SE | Z | $p$ | Mediation % |
|---|---|---|---|---|---|---|
| Indirect | a × b | 7.594 | 0.00237 | 0.320 | 0.749 | 0.0279 |
| Direct | c | 2.72 | 0.05368 | 50.714 | <0.001 | 99.9721 |
| Total | c + a × b | 2.72 | 0.05366 | 50.750 | <0.001 | 100.0000 |
| SI→MED | a | −0.00239 | 0.00390 | −0.613 | 0.540 | |
| MED→PLD | b | −0.31777 | 0.84730 | −0.375 | 0.708 | |
| SI→PLD | c | 2.72230 | 0.05368 | 50.714 | <0.001 | |

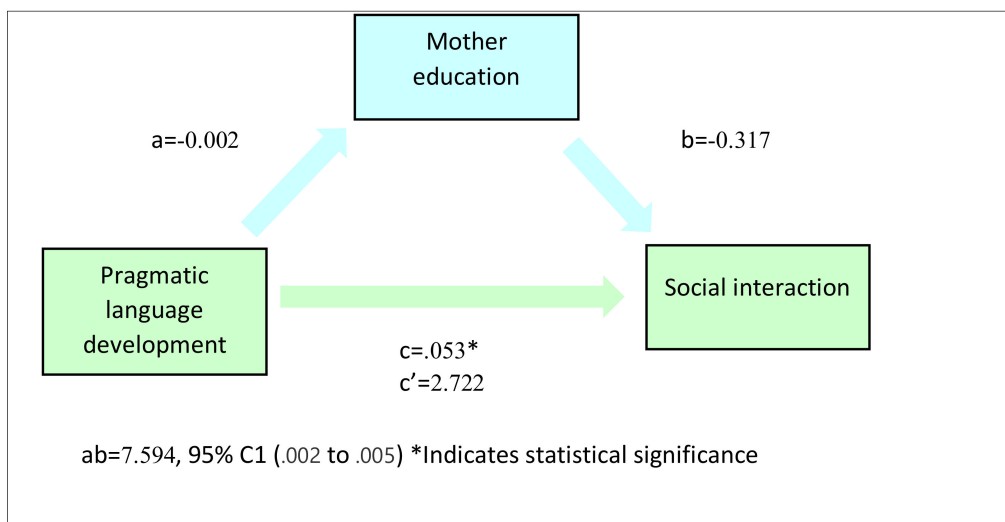

**Figure 15.** Mediation Illustration for Social Interaction, PLD, and Mother Education.

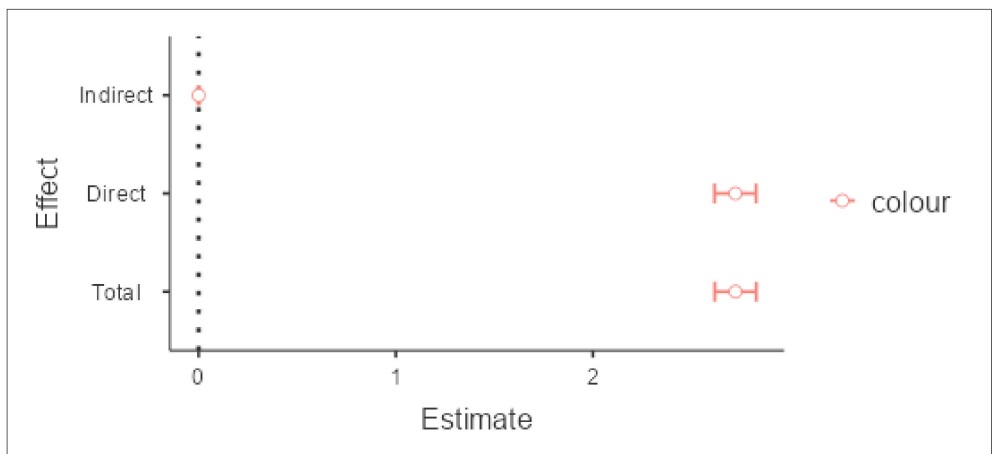

**Figure 16.** Mediation Effect for Social Interaction, PLD, and Mother Education.

### 3.6. Personal Interaction, Pragmatic Language Development, and Parental Employment

The ninth mediation analysis was performed to assess the mediating effect of FE on the linkage between PI and PLD. The results (see Table 10 and Figure 17) reveal that the total effect of PI on PLD was significant (H1: $\beta$ = 2.69, t = 34.848, $p$ < 0.001). With the inclusion of the mediating variable (FE), the impact of PI on PLD was still found significant ($\beta$ = 2.69, t = 34.76, $p$ < 0.001). However, the indirect effect of PI on PLD through FE was found insignificant ($\beta$ = 0.002, t = 0.479, $p$ < 0.632) (see Figure 18). These results indicate that the first criterion for mediation was satisfied but not the last three criteria. Given this, the relationship between PI and PLD is not mediated by FE, and hence mediation is not approved.

**Table 10.** Mediation Analysis for Personal Interaction, PLD, and Father Employment.

| Effect and Path | Label | Estimate | SE | Z | $p$ | Mediation % |
|---|---|---|---|---|---|---|
| Indirect | a × b | 0.00263 | 0.00549 | 0.479 | 0.632 | 0.0977 |
| Direct | c | 2.69042 | 0.07739 | 34.765 | <0.001 | 99.9023 |
| Total | c + a × b | 2.69305 | 0.07728 | 34.848 | <0.001 | 100.0000 |
| PI→FE | a | −5.644 | 5.544 | −1.019 | 0.308 | |
| FE→PLD | b | −4.66 | 8.5814 | −0.543 | 0.587 | |
| PI→PLD | c | 2.69 | 0.0774 | 34.765 | <0.001 | |

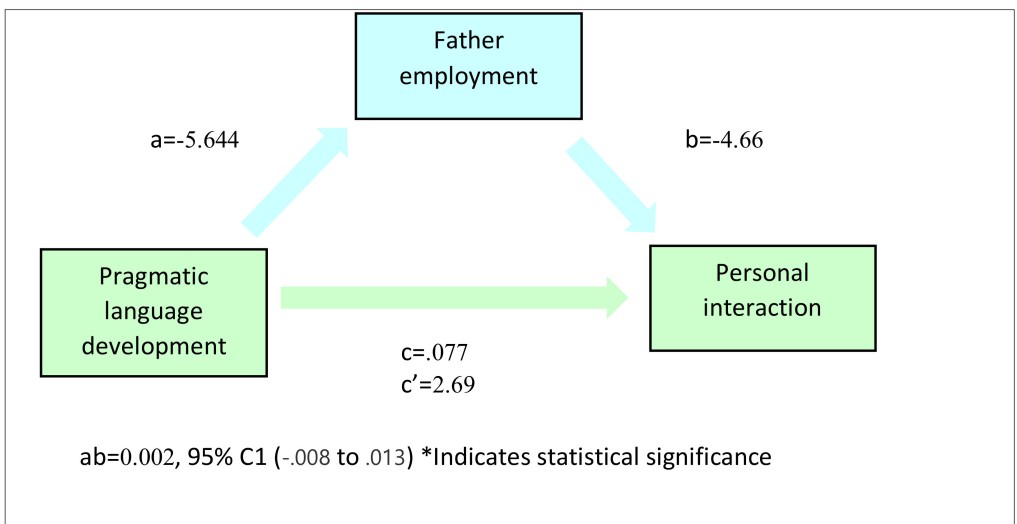

**Figure 17.** Mediation Illustration for Personal Interaction, PLD, and Father Employment.

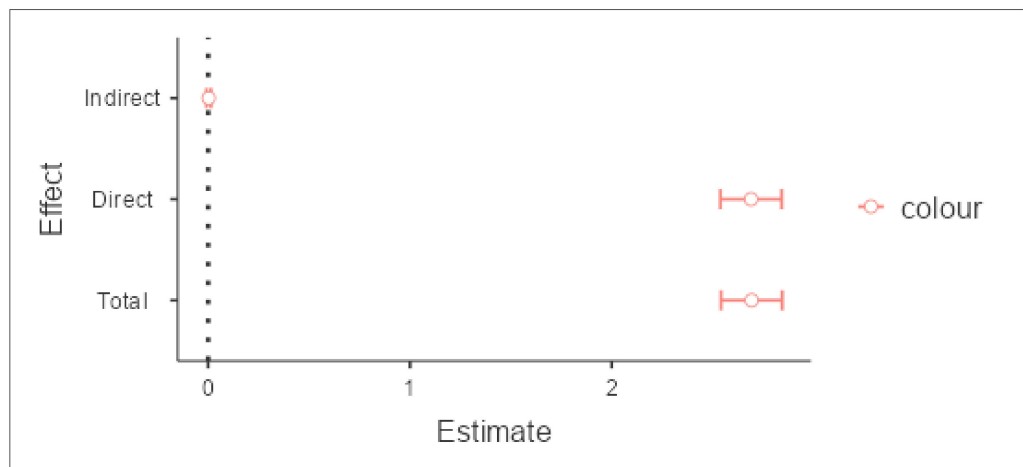

**Figure 18.** Mediation Effect for Personal Interaction, PLD, and Father Employment.

The tenth mediation analysis was performed to assess the mediating effect of ME on the linkage between PI and PLD. The results (see Table 11 and Figure 19) reveal that the total effect of PI on PLD was significant (H1: $\beta$ = 2.69, t = 34.85, $p$ < 0.001). With the inclusion of the mediating variable (ME), the impact of PI on PLD was still found significant ($\beta$ = 2.62, t = 33.10, $p$ < 0.001). The indirect effect of PI on PLD through ME was still found significant ($\beta$ = 0.069, t = 2.59, $p$ < 0.010) (see Figure 20). These results indicate that the four criteria for mediation were satisfied. Given this, the relationship between PI and PLD is partially mediated by ME, and hence *partial mediation is approved*.

**Table 11.** Mediation Analysis for Personal Interaction, PLD, and Mother Employment.

| Effect and Path | Label | Estimate | SE | Z | $p$ | Mediation % |
|---|---|---|---|---|---|---|
| Indirect | a × b | 0.0695 | 0.0268 | 2.59 | 0.010 | 2.58 |
| Direct | c | 2.6235 | 0.0793 | 33.10 | <0.001 | 97.42 |
| Total | c + a × b | 2.6930 | 0.0773 | 34.85 | <0.001 | 100.00 |
| PI→ME | a | −0.00576 | 0.00119 | −4.86 | <0.001 | |
| ME→PLD | b | −12.06322 | 3.94050 | −3.06 | 0.002 | |
| PI→PLD | c | 2.62352 | 0.07927 | 33.10 | <0.001 | |

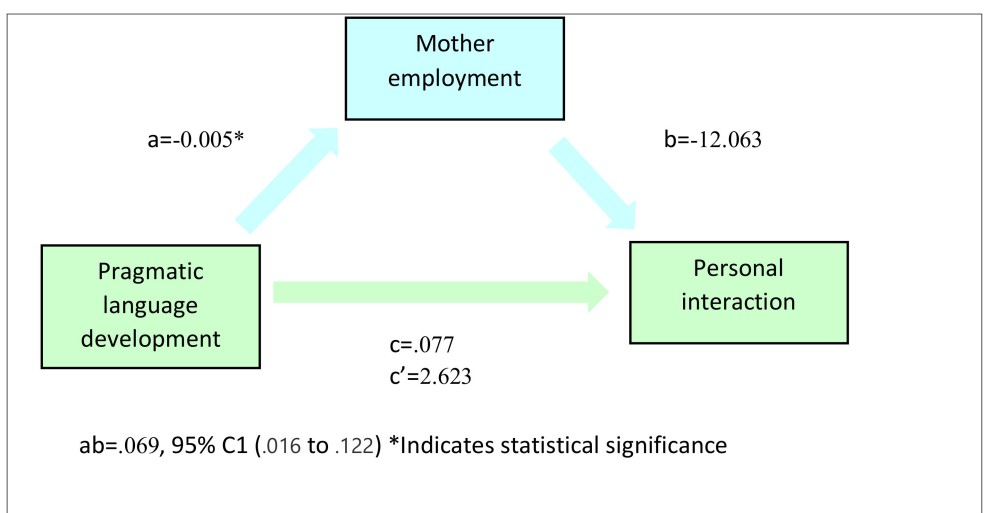

**Figure 19.** Mediation Illustration for Personal Interaction, PLD, and Mother Employment.

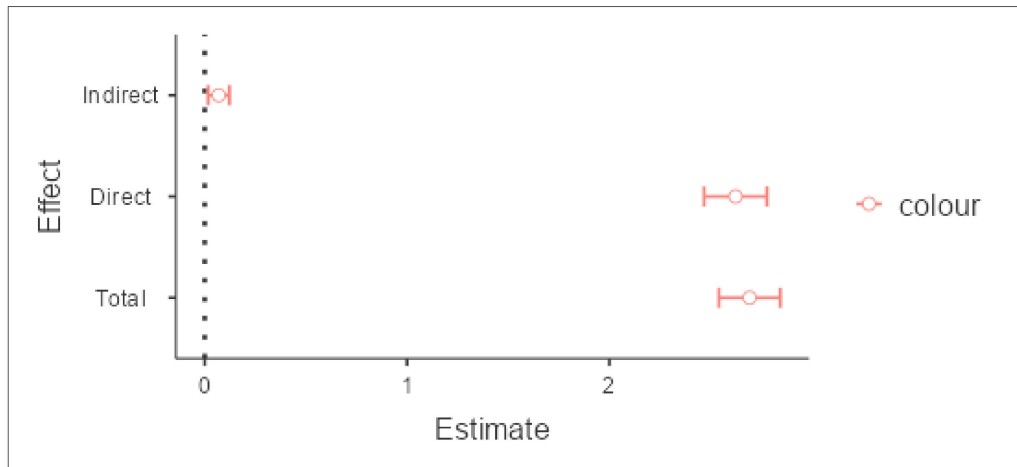

**Figure 20.** Mediation Effect for Personal Interaction, PLD, and Mother Employment.

### 3.7. Personal Interaction, Pragmatic Language Development, and Parental Education

The eleventh mediation analysis was performed to assess the mediating effect of FED on the linkage between PI and PLD. The results (see Table 12 and Figure 21) reveal that the total effect of PI on PLD was significant (H1: $\beta$ = 2.69, t = 34.84, *p < 0.001*). With the inclusion of the mediating variable (FED), the impact of PI on PLD was still found significant ($\beta$ = 2.69, t = 34.90, *p < 0.001*). However, the indirect effect of PI on PLD through FED was found insignificant ($\beta$ = −0.001, t= −0.291, *p < 0.771*) (see Figure 22). These results indicate that the first criterion for mediation was satisfied but not the last three criteria. Given this, the relationship between PI and PLD is not mediated by FED, and mediation is not approved.

**Table 12.** Mediation Analysis for Personal Interaction, PLD, and Father Education.

| Effect and Path | Label | Estimate | SE | Z | p | Mediation % |
|---|---|---|---|---|---|---|
| Indirect | a × b | −0.00124 | 0.00424 | −0.291 | 0.771 | 0.0458 |
| Direct | c | 2.69428 | 0.07719 | 34.904 | <0.001 | 99.9542 |
| Total | c + a × b | 2.69305 | 0.07728 | 34.848 | <0.001 | 100.0000 |
| PI→FED | a | −0.00131 | 0.00421 | −0.310 | 0.756 | |
| FED→PLD | b | 0.94536 | 1.12810 | 0.838 | 0.402 | |
| PI→PLD | c | 2.69428 | 0.07719 | 34.904 | < .001 | |

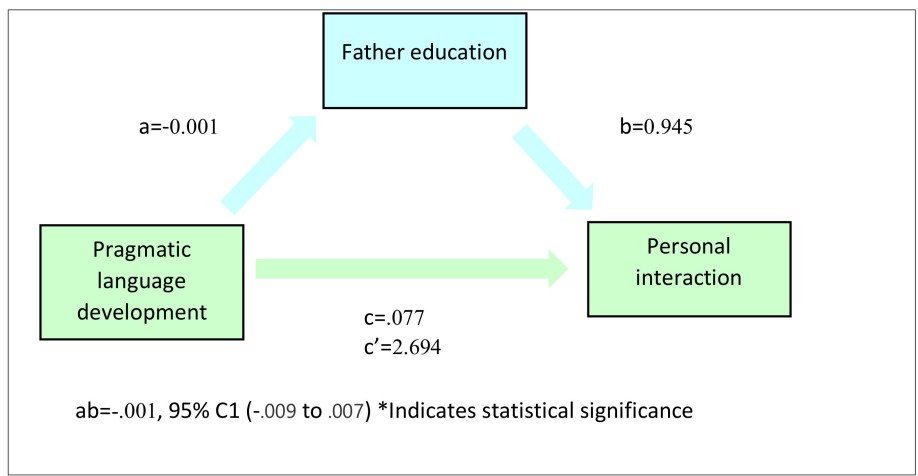

**Figure 21.** Mediation Illustration for Personal Interaction, PLD, and Father Education.

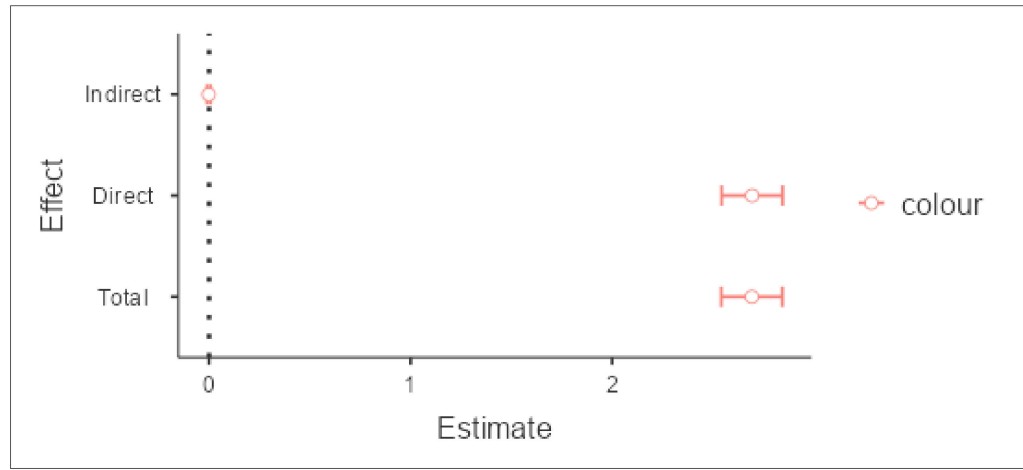

**Figure 22.** Mediation Effect for Personal Interaction, PLD, and Father Education.

The last mediation analysis was performed to assess the mediating effect of MED on the linkage between PI and PLD. The results (see Table 13 and Figure 23) reveal that the total effect of the PI on PLD was significant (H1: $\beta$ = 2.69, t = 34.84, $p$ < 0.001). With the inclusion of the mediating variable (MED), the impact of PI on PLD was still found significant ($\beta$ = 2.69, t = 34.81, $p$ < 0.001). However, the indirect effect of PI on PLD through MED was found insignificant ($\beta$ = −0.001, t = −0.279, $p$ < 0.780) (see Figure 24). These results indicate that the first criterion for mediation was satisfied but not the last three criteria. Given this, the relationship between PI and PLD is not mediated by MED, and mediation is not approved.

**Table 13.** Mediation Analysis for Personal Interaction, PLD, and Mother Education.

| Effect and Path | Label | Estimate | SE | Z | *p* | Mediation % |
|---|---|---|---|---|---|---|
| Indirect | a × b | −0.00127 | 0.00454 | −0.279 | 0.780 | 0.0470 |
| Direct | c | 2.69432 | 0.07739 | 34.816 | <0.001 | 99.9530 |
| Total | c + a × b | 2.69305 | 0.07728 | 34.848 | <0.001 | 100.0000 |
| PI→MED | a | −0.00367 | 0.00405 | −0.907 | 0.364 | |
| MED→PLD | b | 0.34488 | 1.17539 | 0.293 | 0.769 | |
| PI→PLD | c | 2.69432 | 0.07739 | 34.816 | <0.001 | |

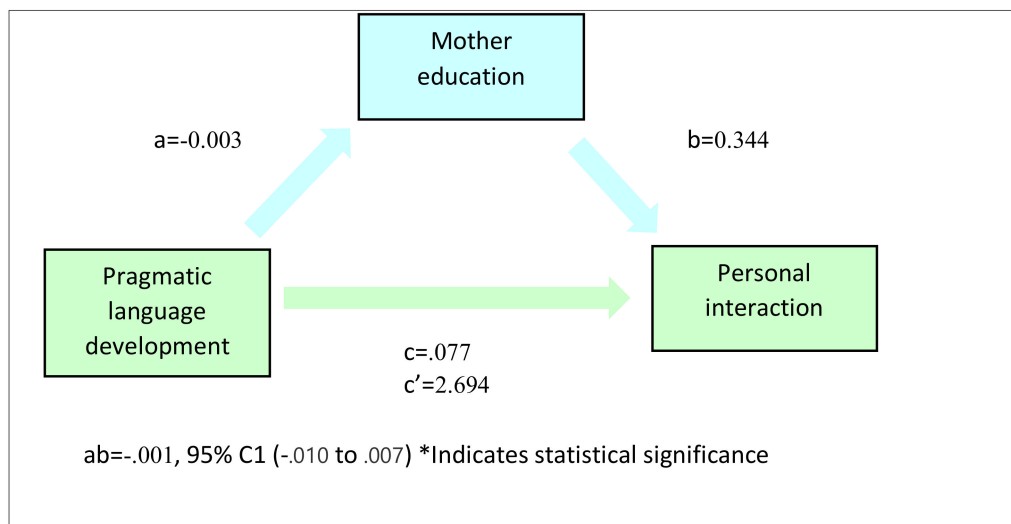

**Figure 23.** Mediation Illustration for Personal Interaction, PLD, and Mother Education.

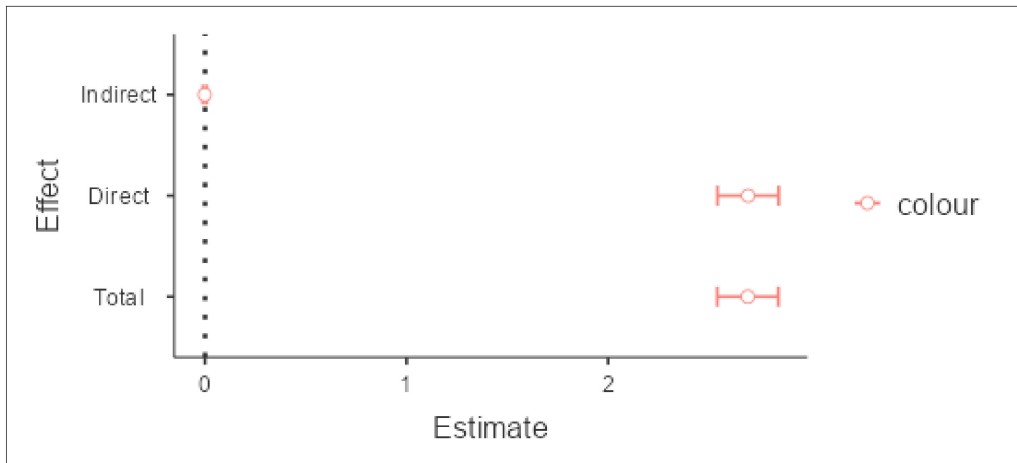

**Figure 24.** Mediation Effect for Personal Interaction, PLD, and Mother Education.

*3.8. Gender and Age as Moderators between PLD and Preschoolers*

3.8.1. Moderation Analysis

Moderation analysis was conducted using Jamovi 2.2.2 to assess if age and gender moderated the relationship between CI, SI, and PI and PLD. In the first step, a simple effect model was created using linear regression with PLD as the outcome variable and CI, SI, and PI as the predictor variables. In the second step, a non-interaction model was created by adding age and gender to the predictor in the linear model in step 1 (simple effects model). In the third step, an interaction model was created by adding the interaction between CI, SI, PI, and age/gender to the predictors in the linear model in step 2 (non-interaction model). In the fourth step, moderation analysis was performed using PLD as the independent variable; CI, SI, and PI as the predictors (independent variables); and age and gender as moderating variables.

3.8.2. Assumptions and Normality

Assumptions for linear regression analysis were conducted for the step 2 and 3 models (non-interaction and interaction models). The assumptions of Durbin–Watson test for auto-correlation ($p < 0.001$), collinearity statistics, and normality test (Shapiro–Wilk) ($p < 0.001$) were assessed. All the models were tested for fit and reported high value (R = 0.95 and $R^2$ = >90) except for PI and age (R = >90 and $R^2$ = >80).

### 3.8.3. Conditions

For moderation to be supported, two conditions must be met. First, the causal predictor variables, CI, SI, and PI, must significantly predict PLD in the simple effect model (step 1). Secondly, the interaction model (step 3) and moderation model (step 4) must explain significantly more variance of PLD than the non-interaction model (step 2). If either of these conditions fail, moderation is not supported.

### 3.8.4. Results

Six moderation analyses were performed for age and gender as moderators; CI, SI, and PI as predictors (independent variables); and PLD as dependent variable. The interactions between CI, SI, PI and age were found to be statistically significant ($\beta$ = −0.151, t = −3.578, $p$ = 0.001; $\beta$ = −0.150, t = −3.20, $p$ = 0.003; $\beta$ = −0.211, t = −3.42, $p$ = 0.001, respectively). Dissimilarly, the interactions between CI, SI, PI, and gender were found statistically insignificant ($\beta$ = 0.133, t = 1.39, $p$ = 0.166; $\beta$ = −0.161, t = −1.477, $p$ = 0.140; $\beta$ = 0.104, t = 0.664, $p$ = 0.507, respectively). The conditional effect of these variables shows corresponding results for the first moderator (age) but not for the second moderator (gender). At average, low, and high moderations, the effect conditions were all effective ($p$ < 0.001). While the interactions were reported statistically insignificant between CI, SI, PI, and gender, the conditional effects indicated moderator relationship at average, low, and high moderations ($p$ < 0.001) (see Table 14 for detailed results).

**Table 14.** Moderation Analysis for Age, Gender, and Pragmatic Language Development.

| Moderation | Estimate | SE | Z | $p$ | Slope | Estimate | SE | Z | $p$ |
|---|---|---|---|---|---|---|---|---|---|
| Classroom interaction | 2.451 | 0.0514 | 47.676 | <0.001 | Average | 2.45 | 0.0523 | 46.9 | <0.001 |
| Age in years | −0.284 | 1.2868 | −0.221 | 0.825 | Low (−1 SD) | 2.61 | 0.0534 | 48.8 | <0.001 |
| Classroom interaction * Age in years | −0.151 | 0.0423 | −3.578 | <0.001 | High (+1 SD) | 2.30 | 0.0803 | 28.6 | <0.001 |
| Classroom interaction * Gender | | | | | | | | | |
| Classroom interaction | 2.498 | 0.0481 | 51.88 | <0.001 | Average | 2.50 | 0.0483 | 51.7 | <0.001 |
| Gender | −5.310 | 2.6804 | −1.98 | 0.048 | Low (−1 SD) | 2.43 | 0.0729 | 33.3 | <0.001 |
| Classroom interaction * Gender | 0.133 | 0.0961 | 1.39 | 0.166 | High (+1 SD) | 2.56 | 0.0627 | 40.9 | <0.001 |
| Social interaction * Age in years | | | | | | | | | |
| Social interaction | 2.683 | 0.0586 | 45.82 | <0.001 | Average | 2.68 | 0.0593 | 45.2 | <0.001 |
| Age in years | −2.364 | 1.3320 | −1.77 | 0.076 | Low (−1 SD) | 2.84 | 0.0585 | 48.5 | <0.001 |
| Social interaction * Age in years | −0.150 | 0.0468 | −3.20 | 0.001 | High (+1 SD) | 2.53 | 0.0911 | 27.8 | <0.001 |
| Social interaction * Gender | | | | | | | | | |
| Social interaction | 2.740 | 0.0550 | 49.834 | <0.001 | Average | 2.74 | 0.0552 | 49.6 | <0.001 |
| Gender | −0.646 | 2.7776 | −0.232 | 0.816 | Low (−1 SD) | 2.82 | 0.0861 | 32.8 | <0.001 |
| Social interaction * Gender | −0.161 | 0.1092 | −1.477 | 0.140 | High (+1 SD) | 2.66 | 0.0678 | 39.2 | <0.001 |
| Personal interaction * Age in years | | | | | | | | | |
| Personal interaction | 2.476 | 0.0763 | 32.44 | <0.001 | Average | 2.48 | 0.0775 | 32.0 | <0.001 |
| Age in years | 9.726 | 1.7211 | 5.65 | <0.001 | Low (−1 SD) | 2.69 | 0.0806 | 33.4 | <0.001 |
| Personal interaction * Age in years | −0.211 | 0.0618 | −3.42 | <0.001 | High (+1 SD) | 2.26 | 0.1170 | 19.3 | <0.001 |
| Personal interaction * Gender | | | | | | | | | |
| Personal interaction | 2.676 | 0.0783 | 34.194 | <0.001 | Average | 2.68 | 0.0783 | 34.2 | <0.001 |
| Gender | −2.634 | 3.8591 | −0.682 | 0.495 | Low (−1 SD) | 2.62 | 0.1193 | 22.0 | <0.001 |
| Personal interaction * Gender | 0.104 | 0.1565 | 0.664 | 0.507 | High (+1 SD) | 2.73 | 0.1007 | 27.1 | <0.001 |

### 3.8.5. Slope Plots

The simple slope plots show that age and gender moderate the relationship between classroom, social, and personal interactions and PLD. In other words, when there are fewer classroom, social, and personal interactions, the level of PLD will be less when age is average, low, and high. On the contrary, when there are more classroom, social,

personal interactions, the level of PLD will be higher when age is average, low, and high (see Figures 25–27). At the same time, when preschoolers tend to be more engaging and engaged during classes, have more relationships in and out of school, and are more conversational, then their PLD is higher regardless of gender (see Figures 28–30). The opposite of these two situations is evidenced in our third objective, where PLD for preschoolers with and without PLI is compared (see Performance Analysis sub-section).

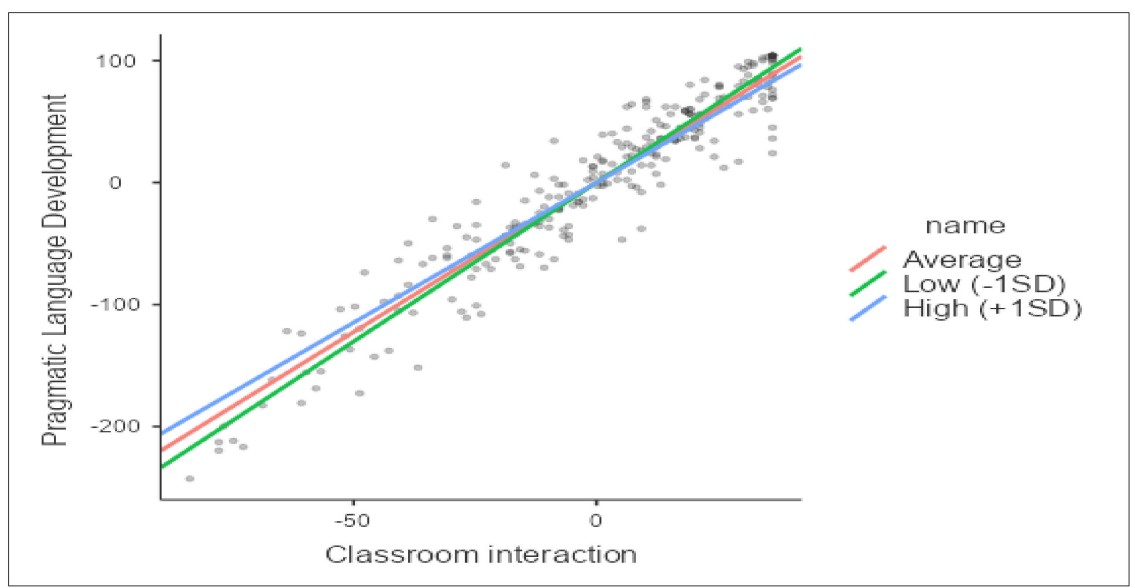

**Figure 25.** Moderation Relationship Direction for Age, PLD, and Classroom Interaction.

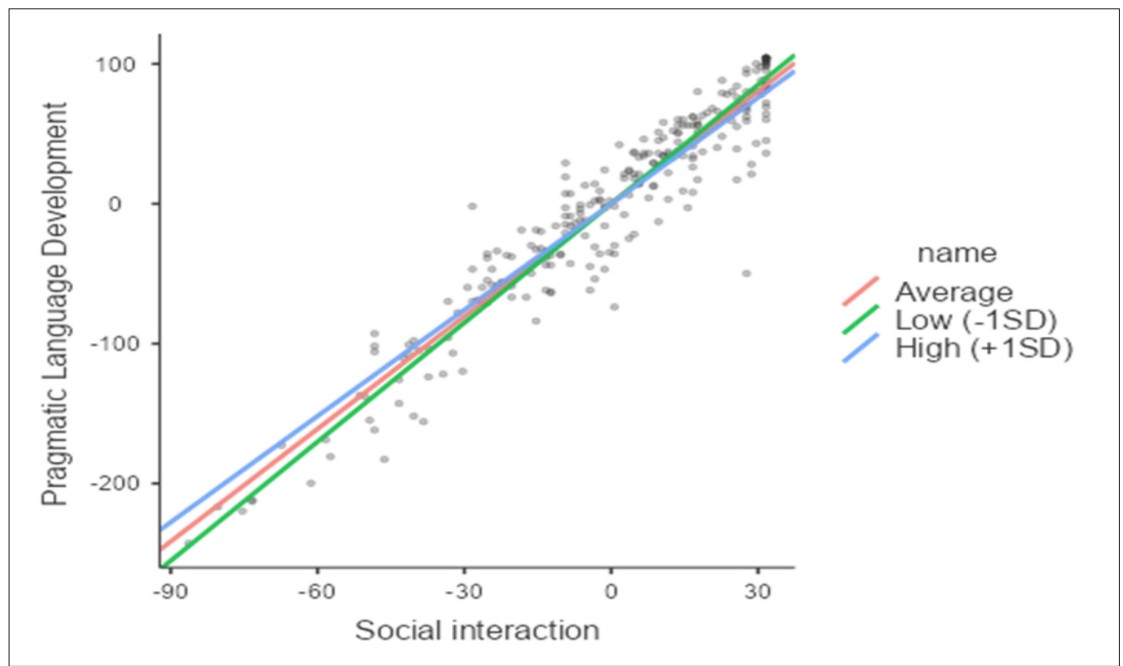

**Figure 26.** Moderation Relationship Direction for Age, PLD, and Social Interaction.

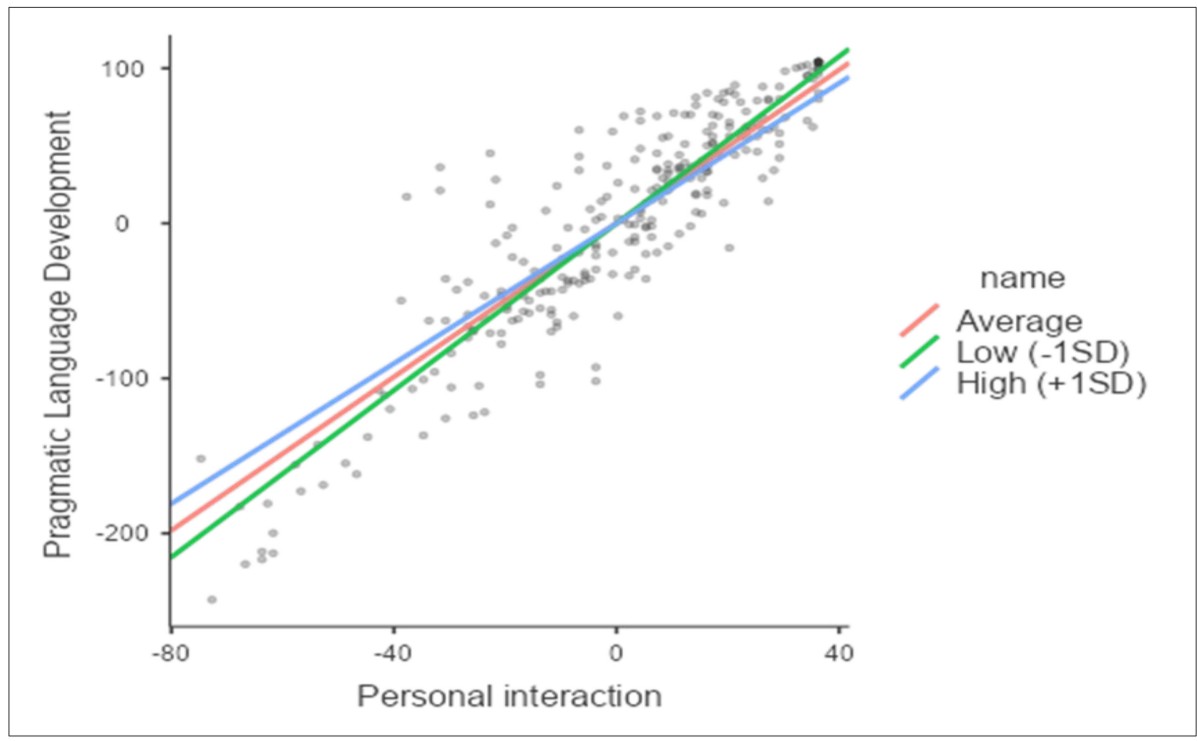

**Figure 27.** Moderation Relationship Direction for Age, PLD, and Personal Interaction.

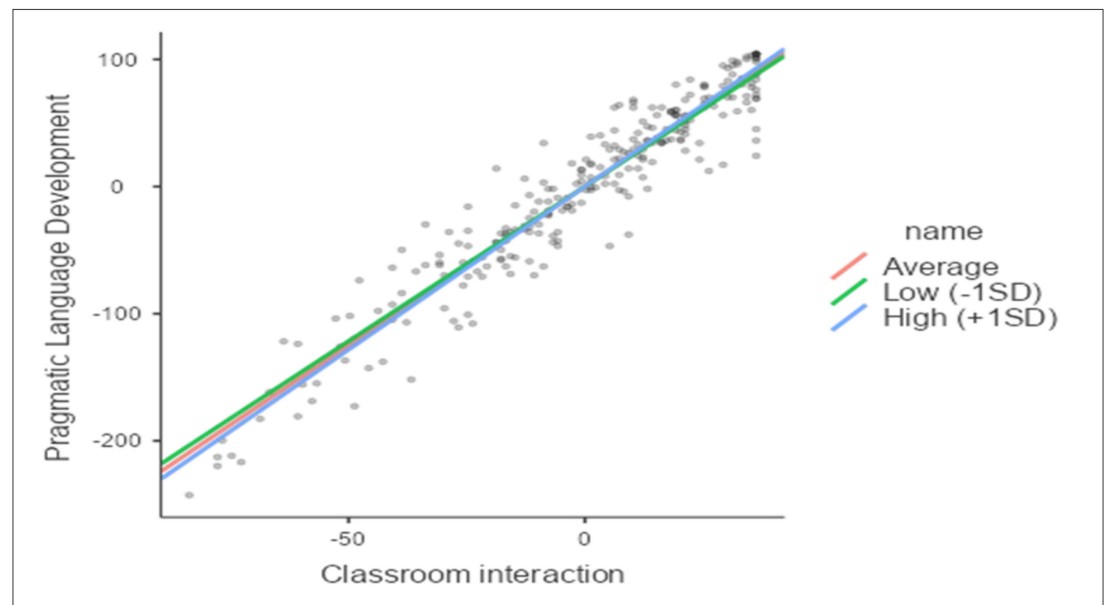

**Figure 28.** Moderation Relationship Direction for Gender, PLD, and Classroom Interaction.

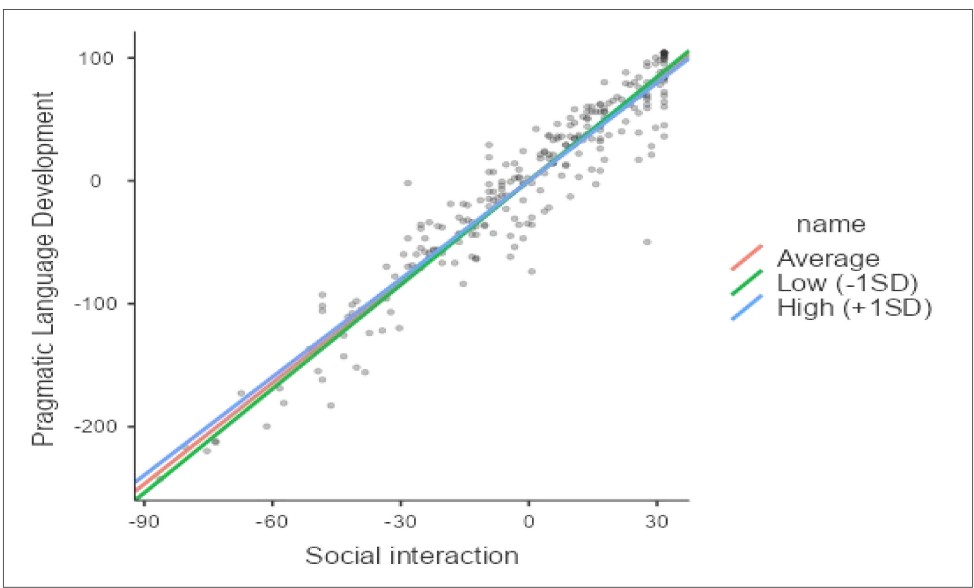

**Figure 29.** Moderation Relationship Direction for Gender, PLD, and Social Interaction.

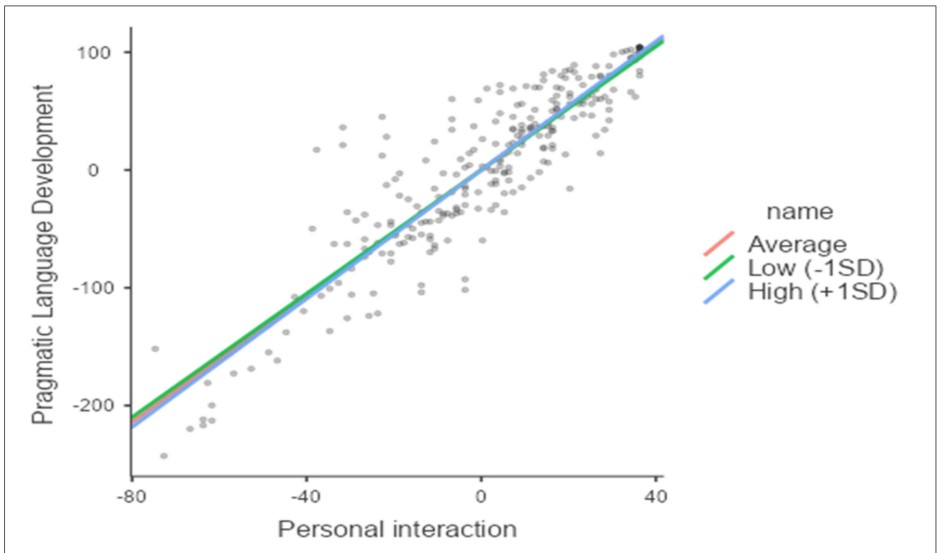

**Figure 30.** Moderation Relationship Direction for Gender, PLD, and Personal Interaction.

*3.9. Pragmatic Language Development Performance in School and Clinical Settings*

The performance of participants was evaluated on the basis of three subscales, namely CI, SI, and PI. Furthermore, these scores are also shown after conversion to standard scores, pragmatic language index, and PLD (i.e., the raw scores for CI, SI, and PI before conversion). The performance of the participants in the school-based group was generally better than that of the clinical group (see Table 15 for detailed statistical differences). Figure 31 illustrates the results of the participants in the two groups after converting the scores in the pragmatic language index to the pragmatic language skills. Participants in the school setting group are among those in the "very superior, superior, and above average" levels. Compared to this, most of the participants in the clinical setting group are in the categories of 'very poor, poor, and below average'.

**Table 15.** Comparison of PLD in preschoolers with and without PLI.

| Variables | N | Mean | SD | Min. | Max. | SE | *p* |
|---|---|---|---|---|---|---|---|
| Age in years—S | 237 | 6.43 | 0.873 | 4 | 7 | | |
| Age in years—C | 27 | 4.59 | 0.747 | 4 | 6 | | |
| Classroom interaction—S | | 104.32 | 22.346 | 30 | 135 | 1.4515 | <0.001 |
| Classroom interaction—C | | 49.81 | 25.391 | 15 | 112 | 4.8865 | <0.001 |
| Standard score CI—S | | 12.97 | 2.750 | 3 | 17 | 0.1786 | <0.001 |
| Standard score CI—C | | 6.93 | 2.999 | 2 | 14 | 0.5772 | <0.001 |
| Social interaction—S | | 108.02 | 21.166 | 36 | 135 | 1.3749 | <0.001 |
| Social interaction—C | | 62.33 | 27.554 | 17 | 120 | 5.3028 | <0.001 |
| Standard score SI—S | | 12.47 | 2.716 | 5 | 17 | 0.1764 | <0.001 |
| Standard score SI—C | | 7.63 | 3.224 | 1 | 15 | 0.6205 | <0.001 |
| Personal interaction—S | | 102.64 | 21.219 | 24 | 135 | 1.3783 | <0.001 |
| Personal interaction—C | | 64.30 | 28.510 | 26 | 122 | 5.4868 | <0.001 |
| Standard score PI—S | | 12.42 | 3.008 | 2 | 18 | 0.1954 | <0.001 |
| Standard score PI—C | | 7.48 | 3.945 | 1 | 16 | 0.7593 | <0.001 |
| Standard score sum—S | | 37.86 | 7.702 | 10 | 52 | 0.5003 | <0.001 |
| Standard score sum—C | | 22.04 | 9.630 | 4 | 45 | 1.8532 | <0.001 |
| Pragmatic Language Development—S | | 314.98 | 58.863 | 118 | 405 | 3.8236 | <0.001 |
| Pragmatic Language Development—C | | 176.44 | 77.560 | 58 | 348 | 14.9264 | <0.001 |
| Pragmatic Language Index—S | | 113.42 | 13.064 | 66 | 137 | 0.8486 | <0.001 |
| Pragmatic Language Index—C | | 86.26 | 16.564 | 56 | 125 | 3.1877 | <0.001 |

Note: S = School setting. C = Clinical setting.

**Figure 31.** Performance of Preschoolers with and without PLI in PLD.

## 4. Discussion

The purposes of this study were threefold: (1) To examine impact of CI, SI, and PI on PLD as mediated by the SES of parents (i.e., employment and education variables). It

was hypothesized that SES will mediate this relationship. A series of regression analyses were carried out to test this hypothesis. (2) To examine the relationship between CI, SI, and PI and PLD and to investigate if age and/or gender act as moderating variables in this relationship. It was hypothesized that CI, SI, and PI will significantly relate to PLD. Additionally, it was predicted that age and/or gender will moderate this relationship, in the sense that they will enhance the level of PLD triggered by CI, SI, and/or PI. A series of moderation analyses were performed. (3) To measure the performance of the preschoolers in PLD and compare performance between children with and without PLI in school and clinical settings. There are three key findings of the present research.

First, the results show that CI, SI, and PI affect PLD directly without the mediation of tested indirect variables, that is, parental employment and parental education. There was only one exception, which was a partial mediation effect approved for the indirect effect of PI on PLD through ME. These findings provide some evidence that preschoolers who tend to be more interactive during school time, socialize more, and build more communication with others are more likely to have a higher PLD. Although this is not mediated by the SES of parents, the evidence showed a direct effect between PLD and CI, SI, and PI. Nevertheless, low interaction, less socializing, and few or no communication skills contribute to a delay or poor PLD in preschoolers. These initial findings are furthered in the discussion section below.

Second, the results show a positive relationship in the case of age but not gender. However, further analysis of conditional effect indicated that both age and gender are moderating variables between CI, SI, and PI and PLD. Above all, this moderation is unconditional and remains active at low, average, and high levels of moderation. A higher level of PLD will be evident when there are more classroom, social, and personal interactions among students of varied ages. At the same time, when preschoolers show more engagement and involvement during class, establish more relationships in and out of school, and display greater conversational skills, their PLD is higher; regardless of gender.

Thirdly, health has a major impact on the typical PLD of preschoolers, one that is very important in addition to the effects of SES, age, and gender. The results of this study provide supporting evidence that preschool children with neurodevelopmental disorders perform much more poorly than their peers without such a history.

These results represent the first direct demonstration of the possible effect of not only SES but its sub-variables (i.e., father employment/education, mother employment/education). However, the overall results are consistent with previous research reporting the interrelation between SES and low and high language development [12,17,27–29]. In line with the findings of the study, Ginsborg [30] reported that children with low socioeconomic status are more likely to experience language delay than children from high socioeconomic status.

The study found that Arabic PLD increases with age; when children grow, they acquire more pragmatic skills. Similarly, there was a pragmatic growth with age found in Italian children's data [31] and in the speech of Norwegian children [31]. Whereas past researchers have found gender differences in language development where girls perform better than boys [32–37], the present study has shown that gender seems to have minor or zero effect on pragmatic language skills of preschoolers in the test's Arabian context. This finding is consistent with several studies in the literature where there is no gender effect [31,38–40]. Some studies have reported that girls performed better than boys in communicative and conversational skills but not other linguistic elements [41,42]. It is found that girls scored higher than boys at the early stage, and the gender differences decrease with the increase in age [43]. Our study confirms that PLD is influenced by factors enhancing PLD (i.e., classroom interaction, social interaction, and personal interaction) and SES factors (i.e., parental education and employment status), although SES showed minor or no differences.

## 5. Implications for Practice

These data have some potential implications. As a first point, the norm that the higher the SES, the greater the chances for typical PLD may be disputed. In light of the evidence

presented, we believe that families with fewer responsibilities tend to spend more time with their children and provide them with more opportunities for language exposure during the early years of life. Secondly, parental education is helpful for providing home education and choosing more practical materials to ensure more exposure to language learning, but it is less influential during infancy and early childhood. In particular, mothers, as well as parents in general, who are more educated tend to be more formal and less comfortable using motherese or *parentese* language with their infants.

## 6. Limitations

Certain limitations of this study could be addressed in future research. The measurement of PLD, for example, may be validated through the use of multiple assessment instruments or through having multiple individuals conduct the same assessment (e.g., parents, teachers, speech-language clinicians). Second, it may be possible to compare SES, age, and gender of groups of children using a matched-group design to learn whether there are differences between the groups. Thirdly, as the context and SES of parents are markedly different among the 22 Arab countries, context could also be compared, specifically for income and welfare effects on children's typical PLD.

## 7. Conclusions

This research examined SES in light of parental employment and education as mediating factors in determining PLD in Saudi Arabian preschoolers both with and without PLI. Moreover, it aimed to examine the moderation effect of PLD with respect to CI, SI, PI, age, and gender. The study also compared characteristics of children's PLD in school settings and clinical settings. We have demonstrated that preschool children who have greater levels of CI, SI, and PI tend to have a higher level of typical PLD regardless of the parents' employment or education, except for the case of mother employment. In addition, we obtained evidence that CI, SI, and PI increase with age, i.e., they grow simultaneously. Nonetheless, these three elements (namely, CI, SI, and PI) do not appear to be moderated or to alter with respect to gender. A further support for this idea may come from finding that typical PLD is attainable when preschool children are manifesting typical mental and physical development, in contrast to children with psychiatric histories who demonstrated atypical PLD. Together our findings suggest that preschool children with more CI, SI, and PI will exhibit more typical PLD. With the accomplishment of this goal, a sustainable society for children is established.

**Author Contributions:** Conceptualization, A.A.; data curation, A.A.; formal analysis, A.A.; funding acquisition, H.A.; investigation, A.A., N.A. and H.A.; methodology, A.A. and L.A.A.; project administration, A.A.; resources, F.Q., H.A., N.A. and L.A.A.; software, A.A.; supervision, A.A.; validation, A.A.; visualization, A.A.; writing—original draft, F.Q. and A.A.; writing—review and editing, A.A and H.A. All authors have read and agreed to the published version of the manuscript.

**Funding:** The research was funded by King Saud University, Riyadh, Saudi Arabia, under the research project RSP-2021/251.

**Institutional Review Board Statement:** All subjects gave their informed consent for inclusion before they participated in the study. The study was conducted in accordance with the Declaration of Helsinki, and the protocol was approved by the Ethics Committee of the Jeddah Institute for Speech and Hearing and Medical Rehabilitation (JISH), Jeddah, Saudi Arabia on 15 November 2021, and College of Education, King Saud University, Saudi Arabia on 10 Ocotber 2021 (RSP-2021/251).

**Informed Consent Statement:** Informed consent was obtained from all subjects involved in the study.

**Data Availability Statement:** The data presented in this study are available on request from the corresponding authors. The data are not publicly available due to ethical restrictions.

**Acknowledgments:** Thank you to all of the preschools that permitted access to their classes and to the teachers who assisted with collecting data. In addition, we would like to thank Jeddah Institute

for Speech and Hearing and Medical Rehabilitation, Jeddah, Saudi Arabia, for collaborating with us to collect data from children with disabilities.

**Conflicts of Interest:** The authors declare no conflict of interest.

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
