# Peer review of "A Study on the Relationship between Pragmatic Language Development and Socioeconomic Status in Arab Preschoolers with and without Pragmatic Language Impairment"

_sustainability, doi:10.3390/su14106369_

Round 1
Reviewer 1 Report
Lines 43-56. There are some variables that may benefit the approach expressed in these lines. Reference to the mother tongue of pupils, socio-economic status and the presence (or not) of pupils with communication or learning difficulties may contribute significantly to better contextualise the initial framework of the article.
Line 72. Change "Hoff, Laursen & Tardif, 2002" to "Hoff et al., 2002".
Line 83. Replace "Betancourt, Brodsky, & Hurt" with "Betancourt et al."
Lines 91 to 96. The bibliographic contextualisation presents works from 20-30 years ago. In order for the research to be up-to-date and take into account the most recent bibliographical production, it is necessary to include bibliographical references published in the last 5 years. In order to justify this need, the conclusions refer to current Italian studies (lines 517 to 533) where recent and current studies are shown.
Lines 160 to 163. Assertions are made without scientific and bibliographical support. This selection of criteria for analysis requires support from previous publications and studies. Otherwise, a generalisation is made that detracts from the subsequent phase of the study.
Figures 1, 2, 5, 7, 9, 9, 11, 13, 13, 15, 17, 19, 21, 23 must fit within the page margins.
Figures 25, 26, 27, 27, 28, 29, 30 appear distorted (narrower in the vertical plane than the original image). If these images are resized, it contributes to the readability of the texts inside each of the margins of the figures.
Author Response
Detailed reply attached.

Reviewer 2 Report
The research problem is well defined and justified; the article is cohesive. A criticism, however, is that the research questions do not appear at the end of Introduction. This is a bit confusing.
For your Conclusions, please consider these questions:
What in the theoretical framework you present in your article do you find the most relevant in the light of your empirical results?
Which of the finding(s) do you find most surprising?
What does your study add to the already existing literature in this field?
Conclusion section should present the results of the study in light of the research questions and theories laid out it the theoretical part of the article.
Round 2
Reviewer 2 Report
The revised version is fine. I would recommend to make a final proofreading.